# Multivessel Coronary Artery Disease Complicated by Diabetes Mellitus Has a Relatively Small Effect on Endothelial and Lipoprotein Lipases Expression in the Human Atrial Myocardium and Coronary Perivascular Adipose Tissue

**DOI:** 10.3390/ijms241713552

**Published:** 2023-08-31

**Authors:** Małgorzata Knapp, Bartłomiej Łukaszuk, Anna Lisowska, Tomasz Hirnle, Jan Górski, Adrian Chabowski, Agnieszka Mikłosz

**Affiliations:** 1Department of Cardiology, Medical University of Bialystok, 15-089 Bialystok, Poland; malgo33@interia.pl (M.K.); anlila@poczta.onet.pl (A.L.); hirnlet@wp.pl (T.H.); 2Department of Physiology, Medical University of Bialystok, Mickiewicza 2C Street, 15-222 Bialystok, Poland; bartlomiej.lukaszuk@umb.edu.pl (B.L.); adrian@umb.edu.pl (A.C.); 3Faculty of Health Sciences, University of Lomza, 18-400 Lomza, Poland; gorski@umb.edu.pl

**Keywords:** apolipoproteins, coronary artery disease, diabetes mellitus, endothelial lipase, heart, lipoprotein lipase, right atrial appendage, coronary perivascular adipose tissue

## Abstract

Endothelial (EL) and lipoprotein (LPL) lipases are enzymes involved in lipoproteins metabolism and formation of atherosclerosis, a pathological feature of coronary artery disease (CAD). This paper examines the role of the lipases in the right atrial appendage (RAA) and coronary perivascular adipose tissue (PVAT) of patients with CAD alone or with accompanying diabetes. Additionally, correlation analysis for plasma concentration of the lipases, apolipoproteins (ApoA-ApoJ) and blood lipids (Chol, HDL-C, LDL-C, TAG) was performed. We observed that CAD had little effect on the lipases gene/protein levels in the RAA, while their transcript content was elevated in the PVAT of diabetic CAD patients. Interestingly, the RAA was characterized by higher expression of EL/LPL (EL: +1-fold for mRNA, +5-fold for protein; LPL: +2.8-fold for mRNA, +12-fold for protein) compared to PVAT. Furthermore, ApoA1 plasma concentration was decreased, whereas ApoC1 and ApoH were increased in the patients with CAD and/or diabetes. The concentrations of ApoC3 and ApoD were strongly positively correlated with TAG content in the blood, and the same was true for ApoB with respect to LDL-C and total cholesterol. Although plasma concentrations of EL/LPL were elevated in the patients with diabetes, CAD alone had little effect on blood, myocardial and perivascular fat expression of the lipases.

## 1. Introduction

In a physiological state, fatty acids are the primary source of energy for the myocardium. The remaining portion of the energy requirement is covered by the oxidation of glucose, lactate and ketone bodies. Fatty acids are delivered to the heart in a different manner, i.e., by the hydrolysis of circulating triacylglycerol (TAG), TAG stored inside of cardiomyocytes and via the uptake of free fatty acids from blood plasma [1]. Although triacylglycerol is essential for myocardial contractility, an excessive plasma TAG level is an independent risk factor for the development of coronary artery disease (CAD) [2]. This cardiovascular disorder is characterized by pathological remodeling of the coronary arterial tree that results in the development of the blood vessels atherosclerosis and subsequent myocardial ischemia. Atherosclerosis is a complex, multifactorial disease involving impaired lipoprotein metabolism, inflammation and changes in the functioning of many cell types present in the vascular wall. Clinically, CAD manifests by stable or unstable angina, myocardial infarction (MI) or sudden cardiac death, which makes it one of the leading causes of death worldwide [3]. To date, numerous risk factors such as type 2 diabetes mellitus (T2DM), obesity, dyslipidemia or hypertension have been associated with an increased incidence of CAD [4,5]. For instance, literature data indicate that the disturbance of glucose metabolism found in diabetes negatively affects the cardiovascular system of patients with established coronary artery disease via the induction of macrovascular and microvascular complications [6]. On the other hand, hyperglycemia itself may be an independent cause of endothelial dysfunction since it triggers the production of protein glycation end products and activates oxidative stress [7].

Plasma TAG level is regulated mostly by the action of enzymes of the triacylglycerol lipase family. Its member, lipoprotein lipase (LPL), is a central player involved in the hydrolysis of TAG-rich lipoproteins. LPL is produced by parenchymal cells such as adipocytes or myocytes and is subsequently transported by GPIHBP1 (glycosylphosphatidylinositol-anchored high-density lipoprotein-binding protein 1) to the luminal surface of the capillary endothelium, where the protein (LPL) performs its action [8]. Importantly, LPL is a rate-limiting enzyme for blood plasma TAG clearance and provision of the released fatty acids to tissue according to its requirement. The activity of LPL is tightly regulated, mainly through post-translational mechanisms, by extracellular proteins. The above-mentioned are divided into two groups: apolipoproteins (e.g., APOC1, APOC2, APOC3 or APOE5) and angiopoietin-like proteins [9]. In clinical practice, plasma LPL concentration is a useful biomarker for a few cardiovascular diseases. Hitsumoto et al., for instance, observed that men with coronary atherosclerosis had a markedly reduced serum LPL level when compared to healthy individuals [10]. Furthermore, it has been shown that LPL may play an important role in the accumulation of lipids inside macrophage cells, which is a relevant step in atheromatous plaque build-up [11].

Roughly 20 years ago, another representative of the lipase gene family was discovered [12]. The enzyme, endothelial lipase (EL), is distinct from the other members of the group because it is synthetized mainly by the endothelium itself and to a lesser extent by macrophages and smooth muscle cells of the arterial wall [13,14]. In contrast to LPL, EL displays a significant phospholipase activity which enables it to hydrolyze HDL-phospholipids at the sn-1 position; moreover, it has a much lower affinity for triacylglycerol and cholesterol esters present in HDL particles [13,15,16]. Importantly, low plasma HDL-C (high-density lipoprotein cholesterol) level is considered to be an important risk factor for the development of atherosclerotic cardiovascular diseases (CVDs) [17]. Despite that established belief, recent research has redefined the relationship between HDL-C concentration and CVD risk to be rather a “U-shaped” curve and not a straight line. Based on that relatively new finding, the optimal HDL-C level in blood plasma is believed to lie within the range of 40–70 mg/dL for men and 50–70 mg/dL for women. It has been shown that individuals with very low (<40 mg/dL for men and <50 mg/dL for women) and very high (>80 to 90 mg/dL) HDL-C levels have a greater risk of CVD-related mortality compared to those with HDL-C levels within the intermediate ranges [18,19]. There are several reports suggesting the existence of an inverse relationship between blood plasma EL concentration/activity and HDL-C level. In both in vitro and animal models, EL overexpression is accompanied by a decrease in plasma HDL-C [20], whereas EL deficiency [16] or its inactivation [21] can lead to an increase in the lipoprotein level. Similarly, in humans, plasma EL level inversely correlates with HDL-C concentration but positively correlates with the occurrence of atherosclerosis and metabolic syndrome [22]. For example, higher EL activity in individuals with severe obesity was associated with lower levels of HDL-C and apolipoprotein A-I [23]. Similarly, EL serum level was elevated in the diabetic patients treated with oral antidiabetic drugs but did not change in the patients on insulin therapy [24]. In addition, in vitro experiments showed that insulin dose-dependently decreased EL transcript and protein content in human aortic endothelial cells [24]. Since an elevated HDL-C concentration is considered to be one of the main factors protecting against the development of atherosclerosis, EL may be a potential pro-atherogenic factor in humans. In line with that notion, it has been reported that human plasma EL concentration was associated with subclinical atherosclerosis [22] and might be considered as an independent risk factor for coronary artery disease [25,26]. Nonetheless, another study found no association between plasma EL level and the severity of coronary artery disease [27]. In addition to the level of HDL-C, “HDL particle functionality” defined by the content of apolipoproteins and lipids in HDL-C seems to be a far more important factor that possibly protects against atherosclerosis. EL modifies the structural and functional properties of HDL-C. Therefore, it reduces its role in reverse cholesterol transport and its anti-atherosclerotic properties [28]. Nevertheless, the literature data indicate that the effect of EL on coronary atherosclerosis is more complex than initially expected.

Generally, it is believed that the pathogenesis of atherosclerosis begins with damage to the tunica intima of blood vessels; therefore, endothelial cells are considered to be the main therapeutic target of the disease [29]. Interestingly, recent studies indicate the potential involvement of perivascular adipose tissue (PVAT) in the pathogenesis of coronary atherosclerosis [30]. Physiologically PVAT contributes to the maintenance of vascular homeostasis, but in pathological conditions, it releases a set of pro-inflammatory cytokines, stimulates oxidative stress and alters macrophage phenotype; these, in turn, lead to microcirculatory inflammation, endothelial dysfunction, accelerated atherosclerosis and ultimately development of CAD [31]. However, little is known about the expression and function of endothelial and lipoprotein lipases in the human myocardium and perivascular adipose tissue in the course of CAD.

The influence of the triacylglycerol lipases on multivessel coronary artery disease has not been fully elucidated, most likely because LPL and EL may exert both anti-atherosclerotic and pro-atherosclerotic effects. Since PVAT may interact locally with the myocardium and coronary arteries, we assessed the expression of both the lipases in the human right atrial appendage and perivascular adipose tissue of patients with coronary artery disease alone and with accompanying diabetes. In addition, plasma EL and LPL concentrations, as well as apolipoproteins levels, were measured and the association between these markers and CAD or diabetes was also established.

## 2. Results

### 2.1. Basic Characteristics

The basic clinical characteristics of the patients enrolled in the study are given in Table 1. In general, there were only a few between-group differences since most of the patients were treated with statins. Still, the diabetic subjects had a higher level of fasting blood glucose than their non-diabetic counterparts (+21% for NCAD(+) vs. NCAD(−) and +41% for CAD(+) vs. CAD(−), *p* < 0.05). Additionally, the subjects in the CAD(+) group had lower HDL-C blood concentration when compared to their counterparts from the NCAD(−) group (−22%, *p* < 0.05). Naturally, the groups with diabetes (NCAD(+) and CAD(+)) were characterized by the presence of this condition. In turn, the patients with coronary artery disease had a greater frequency of a past myocardial infraction (~38% in CAD(−) and ~30% in CAD(+), *p* < 0.05) than the control subjects (0% for both NCAD(−) and NCAD(+)). Surprisingly, we did not find any significant between-group differences with regard to blood lipids concentration as the levels of TAG, Chol, HDL-C and LDL-C were rather stable. However, subsequent analysis demonstrated a strong positive correlation (r ≈ 0.9, *p* < 0.05) between the plasma level of total cholesterol (Chol) and the cholesterol in low-density lipoproteins (LDL-C). This association was also present (to a different extent) when the correlations were analyzed separately within a group.

### 2.2. Plasma Concentration of EL and LPL

The level of circulating EL in the diabetic patients without CAD was significantly higher than in their counterparts without diabetes (+4.9 fold for NCAD(+) vs. NCAD(−), *p* < 0.05, Figure 1A). Interestingly, the concentration of the enzyme in the patients with CAD, both with and without diabetes (CAD(−) and CAD(+)) did not differ from the one observed in the control group (NCAD(−)). Similarly, with respect to LPL, we observed a statistically significant increase in its plasma concentration in the NCAD(+) group (+40%, *p* < 0.05, Figure 1B). Interestingly, blood concentrations of EL and LPL were positively correlated with each other (r = 0.42, *p* < 0.05) when all the groups were considered together. Moreover, this phenomenon seems to be preserved (to a greater or smaller extent) in each of the groups analyzed separately.

### 2.3. Plasma Concentration of Apolipoproteins

Blood plasma concentrations of the investigated apolipoproteins were relatively stable between the studied groups (Figure 2). The only differences were observed with respect to ApoA1, ApoC1 and ApoH. The concentration of ApoA1 was significantly lower in both the groups with coronary artery disease when compared to the non-CAD patients (−30%, *p* < 0.05; Figure 2A). On the other hand, the concentration of ApoC1 in the group with the heart disease was greater when compared with the control (+42% for CAD(−) vs. NCAD(−), *p* < 0.05; Figure 2D). Similarly, the blood plasma level of ApoH in the patients with coronary artery disease but without diabetes was significantly higher when compared with both the control groups (+21% for CAD(−) vs. NCAD(−) and +35% for CAD(−) vs. NCAD(+), *p* < 0.05, Figure 2H).

The correlation analysis for apolipoproteins and other blood plasma parameters (CRP, EL, LPL, Chol, HDL-C, LDL-C and TAG) displays an interesting pattern (Figure 3A). Looking at all the groups together it appears that the levels of all the analyzed apolipoproteins are strongly positively correlated with each other. The correlation coefficients lay roughly within the r = 0.8 to r = 0.5 range (*p* < 0.05). This pattern is also present in the correlations determined for the groups separately (Figure 3B,D,E). However, it seems to be somewhat less apparent in the control patients with diabetes mellitus (NCAD(+), Figure 3C). Nonetheless, we must point out that the discussed group was significantly less numerous than the other groups and that after adjusting for false positives (BH multiplicity correction) none of the *p*-values (*p* > 0.05) for the correlations depicted in Figure 3C were statistically significant. Moreover, Figure 3A seems to reveal also other patterns, namely, the levels of some apolipoproteins are positively correlated with some blood lipids. For example, the concentration of ApoC3 is strongly positively correlated (r = 0.68, *p* < 0.05) with the amount of TAG in the blood. A similar relationship holds to a different extent in the other three groups (Figure 3B,D,E); however, it reached the statistical significance level only in the case of CAD(+) (r = 0.91, *p* < 0.05). Moreover, the content of ApoD was positively correlated with blood plasma TAG level (Figure 3). This was especially evident when observed collectively (r = 0.39, *p* < 0.05, Figure 3A) and separately in NCAD(−) (r = 0.46, *p* > 0.05, Figure 3B) and CAD(+) (r = 0.59, *p* > 0.05, Figure 3E). Regarding the other apolipoproteins, the blood plasma level of ApoB was positively correlated with the concentration of LDL-C as observed in all the patients together (r = 0.47, *p* < 0.05, Figure 3A) as well as in the subjects without CAD and diabetes (NCAD(−), r = 0.84, *p* < 0.05, Figure 3B). The apolipoprotein B was also positively correlated with total cholesterol which was evident in all the data collectively (r = 0.43, *p* < 0.05, Figure 3A), the control patients (NCAD(−), r = 0.86, *p* < 0.05, Figure 3B) and to a less extent in the other groups (*p* > 0.05).

### 2.4. Tissue Expression of EL and LPL

#### 2.4.1. Right Atrial Appendage

In the human atrial myocardium, mRNA expression of EL was relatively stable between all the analyzed groups (Figure 4A). Similarly, EL protein expression showed only minor fluctuations. Particularly, we found that the protein content was slightly diminished in NCAD(+) (−16%, NCAD(+) vs. NCAD(−), *p* < 0.05) but increased in CAD(−) (+29%, CAD(−) vs. NCAD(+), *p* < 0.05, Figure 4C). Furthermore, the transcript level of LPL also appeared to be relatively stable, with a smaller gene amount found in the patients with coronary artery disease when it was accompanied by diabetes than with the condition alone (−62%, *p* < 0.05, for CAD(+) vs. CAD(−), Figure 4B). In the case of LPL protein content, the only statistically significant difference observed was between NCAD(+) and CAD(−). The latter had a lower amount of LPL (−64%, *p* < 0.05) in the analyzed tissue (Figure 4D).

#### 2.4.2. Perivascular Adipose Tissue

Perivascular adipose tissue was characterized by a greater EL mRNA expression in all the studied groups when compared to NCAD(−) (NCAD(+): +98%; CAD(−): +102%; CAD(+): +155%, *p* < 0.05, Figure 5A). Surprisingly, this was not reflected in the protein level of EL as it was relatively stable among the investigated groups. The examination of mRNA for LPL revealed its greatest abundance in the NCAD(+) group (+132%, NCAD(+) vs. NCAD(−), *p* < 0.05; NCAD(+) vs. CAD(−) +74%, *p* < 0.05; NCAD(+) vs. CAD(+) +164%, *p* < 0.05, Figure 5B). However, the above did not echo in LPL protein expression as it remained relatively stable between the analyzed groups (Figure 5D).

#### 2.4.3. Comparison of Lipases Expression in the Myocardium and PVAT

We also decided to assess the tissue expression of EL and LPL at the transcript and protein levels (Figure 6). Since the myocardial tissue has a significantly higher total protein content than perivascular fat, the amount of mRNA and protein between the two tissue types was normalized to a gene/protein reference level. Lipase gene expression was normalized to RPL13A, while protein expression was normalized to GAPDH. For the comparison we used the samples from the control group, i.e., patients without CAD and without diabetes (NCAD(−) group). Interestingly, we observed a roughly +1-fold greater level of mRNA for EL in the right atrial appendage when compared with perivascular fat (*p* < 0.05, Figure 6A). Whereas in the case of its protein content, an approximately +5-fold difference was observed between the tissues with the greater amount found in the human myocardium (*p* < 0.05, Figure 6C). Regarding the lipoprotein lipase, it seemed to follow a similar pattern, with a vastly greater expression in the right arterial appendage (+2.8 fold for LPL mRNA, *p* < 0.05; +12-fold for LPL protein, *p* < 0.05, Figure 6B,D).

Moreover, we conducted correlation analyses of the obtained data for EL and LPL tissue expression with other blood-born parameters (Figure 7). However, the results diverge significantly between the groups, and we were unable to detect a clearly repeating pattern (Figure 7A–E).

## 3. Discussion

A growing body of evidence suggests that triacylglycerol lipases, i.e., LPL or EL, may influence the onset and progression of atherosclerosis, i.e., an underlying pathological feature of coronary artery disease. Both lipases are not only important enzymes in the metabolism of lipoproteins, but they also participate in the vascular regulation of local immunological inflammatory processes, the processes that are involved in the progression of atherosclerosis. Since its discovery, endothelial lipase has been known as a major determinant of plasma HDL-C concentration [16,32]. Recent data have shown that EL is also a potent modulator of the structural and functional properties of HDL-C [33]. In humans, plasma EL level is positively correlated with the amount of inflammatory biomarkers [34,35]. Since inflammation is firmly associated with HDL-C dysfunction, EL is acknowledged to be a biological factor involved in lipid metabolism and the formation of atherosclerosis. However, the role of EL in the pathogenesis of the condition is not well understood given the apparently contradictory data that have been published so far. In animal models, some reports indicated that EL acts as a pro-atherosclerotic molecule [22,36], whereas other studies postulate its anti-atherosclerotic effect [37] or no effect at all [38]. Nevertheless, human studies revealed that plasma EL concentration/activity is increased in patients with CAD and is inversely related to HDL-C concentration [22,26,34]. However, our results demonstrate that plasma EL level was relatively unchanged in the patients with CAD compared to the non-CAD patients. At the same time, the blood enzyme level was elevated in the NCAD patients with diabetes when compared to their non-diabetic counterparts (75.55 ng/mL vs. 22.52 ng/mL, for NCAD(+) vs. NCAD(−), *p* < 0.05). To some extent, our results appear to be inconsistent with those published by Shiu et al. [24]. The authors found an elevated serum EL concentration in diabetic patients; however, after the start of insulin therapy, its concentration decreased. The likely explanation for the discrepancy is the duration of insulin treatment. The patients reported by Shiu et al. [24] received prolonged hormone treatment, whereas our diabetic patients had been treated with oral anti-diabetic agents, and the treatment was switched to insulin only two days before the surgery, i.e., two days before the blood samples were taken. In addition, we did not find any between-group differences regarding the plasma HDL-C concentration; moreover, its level did not correlate with plasma EL values in the patients with NCAD and CAD. These observations are supported, at least in part, by the findings of Trbušić et al. who reported a lack of correlation between EL and HDL-C levels in the patients with stable coronary artery disease [27]. The absence of changes in HDL-C level might be partially explained by the fact that most of our patients (in every group) were treated with statins. Since, statin therapy can slow down, stop and even reverse the progression of atherosclerotic disease, it certainly helped to maintain the proper level of the lipoprotein in the CAD groups. This is of vital importance since Sasso and co-authors revealed a U-shaped relationship between plasma HDL-C level and cardiovascular complications of diabetes [19]. Briefly, the authors reported that both low (<30 mg/dL) and high (>60 mg/dL) concentrations of HDL-C contribute to the development of diabetic retinopathy, whereas the intermediate levels (like the 40–55 mg/dL range reported in our study, see Table 1) have no such effect [19]. Another beneficial effect of statins on CAD is the reduction of inflammation, as reflected by C-reactive protein levels [39]. Similarly, in our study, we observed a reduction in plasma CRP concentration in patients with CAD after statin therapy. This may upset the relationship between the concentration of circulating endothelial lipase and markers of inflammation. Therefore, although in the literature plasma EL levels are positively associated with CRP [34,35], we found no correlation between the two in our CAD patients. Moreover, CAD is a multifactorial disease and it is unlikely that a single protein (like EL or CRP) triggers its onset, more likely such a protein level reflects the existence of an underlying local pathological process that takes place in the arteries. Therefore, further research is needed to clarify the role of EL in the development of this condition. Furthermore, we found that the patients with T2DM but without CAD had a significantly greater plasma LPL concentration (NCAD(+) vs. NCAD(−)). Previous studies on human subjects showed either an increase [40], a decrease [41] or no change in the circulating LPL level in diabetic patients [42]. Although an association between the circulating LPL level and cardiovascular disease has been postulated, in the present study, we were unable to confirm such a supposition since plasma LPL level was relatively stable and only a trend (*p* > 0.05) towards decreased plasma LPL level in the patients with CAD was noticed (CAD(−) vs. NCAD(−)). Still, other authors observed a lower plasma LPL concentration in patients with CAD, hypertriglyceridemia or metabolic syndrome [10]. Han et al. found that out of the three lipases, i.e., EL, HL and LPL, only the serum levels of the first two were positively correlated with CAD progression. Despite that, only serum HL concentration was considered to be an independent risk factor for CAD progression [43].

In addition, we examined the plasma concentration of nine apolipoproteins (ApoA1, ApoA2, ApoB, ApoC1, ApoC3, ApoD, ApoE, ApoH and ApoJ) in the patients with CAD. Moreover, we also evaluated their association with blood lipid profiles to find potential plasma biomarkers for CAD. The above is of interest since apolipoproteins are constituents of HDL-C and LDL-C [44,45]. The former displays a cardioprotective function [45], whereas the latter is considered to be a risk factor that contributes to major adverse cardiac events (MACEs) [46]. Interestingly, we noticed that the groups with CAD had lower ApoA1 levels when compared to the control patients (CAD(−) and CAD(+) vs. NCAD(−)). This is quite interesting since ApoA1 is a predominant apolipoprotein constituent of HDL particles with a well-documented atheroprotective function [45]. Although we did not observe any differences in blood HDL-C concentration in patients with CAD, the reduced ApoA1 could potentially influence HDL-C functionality and be an early marker of CAD. This seems to be even more likely given that in the available literature data, the deletion of the ApoA1 gene in mice contributed to a greater incidence of atherosclerosis independently of plasma HDL-C content [47]. The opposite was true in the transgenic mice with the human ApoA1 gene overexpression [48]. However, we must be cautious not to draw too far-reaching conclusions as some human clinical trials demonstrated that increased levels of HDL-C and ApoA1 do not always reduce coronary atherosclerosis progression [49]. Moreover, we found ApoA1 to be positively correlated (r ≈ 0.6) with ApoA2 and ApoJ (Figure 3). However, due to the scarcity of literature data, we are unable to explain such an association between the three apolipoproteins in the light of their possible cardioprotective function. We believe future studies in this area are warranted. In the remaining set of apolipoproteins, ApoC1 and ApoH had significantly greater concentrations in patients with CAD (CAD(−) vs. NCAD(−), Figure 2). In addition, the increased ApoC1 content was accompanied by a lower plasma LPL level in CAD(−). Numerous observations indicate that ApoC1 inhibits lipoprotein lipase, a critical determinant of plasma TAG clearance and the uptake of the released fatty acids to tissue [50]. This could potentially explain the elevated plasma TAG level observed in the CAD patients. On the other hand, previous studies suggest that the ubiquitous expression of ApoC1 in various cell types may contribute to the development of atherosclerotic plaque. For instance, ApoC1 transcript level was significantly higher in the carotid and femoral atherosclerotic lesions than in healthy arteries [51]. Nevertheless, it appears that atherogenesis as a multifactorial process is not controlled by a single determinant, but rather by the interplay and balance between different types of apolipoproteins.

### 3.1. Tissue Expression of the EL and LPL in Patients with CAD

#### 3.1.1. Atrial Myocardium

Endothelial lipase is abundantly expressed in the tissues with high metabolic rate and vascularization, including the heart, lungs, kidneys, liver, thyroid or placenta. In addition, in a healthy adult, EL mRNA is highly expressed in the coronary arteries [15]. In the present study, myocardial endothelial lipase transcript content was not affected by multivessel coronary disease or type 2 diabetes, while its protein expression was significantly decreased in the patients qualified for mitral or aortic valve replacement without CAD but with type 2 diabetes mellitus (NCAD(+) group). It is hard to satisfactorily explain this finding since reports on EL expression in the myocardium seem to be missing. Still, Vigelsø et al. reported that EL protein level in the human skeletal muscle was similar between healthy men and men with T2DM or impaired glucose tolerance [52]. However, in the above-mentioned study, the patients were treated with oral hypoglycemic agents or insulin and were not stratified by the type of glucose-lowering drug used. Unfortunately, to the best of our knowledge, there are no other data on the influence of diabetes on EL expression in other muscle tissue.

Lipoprotein lipase (LPL) performs its function at the luminal surface of endothelial cells; however, the expression of the enzyme in the cells is rather low. In the heart, the majority of LPL is synthesized in cardiomyocytes and only then it is transported to the luminal side of the capillary endothelium. In that location, it mediates the metabolism of lipoproteins and their uptake. We noticed a trend towards higher LPL protein content in the myocardium of NCAD(+) (+45% for NCAD(−) vs. NCAD(+), *p* > 0.05), while its transcript level was relatively stable between the groups. In general, in the diabetic heart, the usage of glucose for energy production decreases and the dependence on FA utilization to generate ATP dominates over that observed in the healthy heart. LPL-mediated hydrolysis of circulating TAG was suggested to be the main source of FAs for the diabetic cardiac muscle. Accordingly, it was found that the content of LPL, a rate-limiting enzyme for the clearance of circulating TAG, was increased in rats’ models of acute and moderate diabetes [53]. Similarly, LPL expression in the heart of the rats with streptozotocin-induced diabetes was significantly increased [54]. This increase in LPL content was associated with a series of post-translational modifications that substantially increased the enzyme transfer from the myocyte to the vascular lumen but were not connected with the gene modification. On the other hand, insulin-deficient mice have been shown to have a reduced LPL transcript level in the peripheral tissues, such as the heart, skeletal muscle or adipose tissue [55].

#### 3.1.2. Perivascular Adipose Tissue

Recently, researchers’ attention has been drawn to perivascular fat (PVAT), i.e., a type of adipose tissue that surrounds arteries. PVAT may interact in a paracrine fashion with the myocardium and coronary arteries through the secretion of proinflammatory and proatherogenic adipokines. This could potentially contribute to the development of CAD. In the current study, we found large increases (roughly +100%) in the expression of mRNA for EL in the perivascular adipose tissue of the patients with coronary artery disease (CAD(−) and CAD(+)) and the control patients with diabetes (NCAD(+)) when compared to NCAD(−). Moreover, in this research, protein expression of EL in perivascular adipose tissue was relatively stable across the examined groups (Figure 5). This means that the changes in EL mRNA expression did not reflect the changes in the enzyme protein level. It is hard to explain this finding; however, we must remember that the correlation between the level of mRNA and the amount of its protein is usually weak (or none at all), as the latter is mostly regulated by post-transcriptional mechanisms. In addition, the increase in the level of mRNA for EL in both the CAD groups (CAD(−) and CAD(+)) was similar to the one found in the group without CAD but with T2DM (NCAD(+)). This indicates that atherosclerosis itself contributes to the expression of this compound (since it is usually present in CAD and T2DM).

The content of messenger RNA (mRNA) for LPL was significantly elevated only in the patients without CAD but with T2DM (NCAD(+) group), while its protein expression was relatively stable across all the studied groups. Our results are somewhat consistent with the previous report by Barchuk et al. [56]. The authors found no differences in the expression of the gene for LPL and its protein in the patients undergoing coronary artery bypass graft or valve replacement complicated by diabetes in epicardial or subcutaneous adipose tissue [56]. Nevertheless, we must remember that this is all circumstantial evidence; as to the best of our knowledge, the expression of this enzyme has not been previously reported in PVAT.

### 3.2. Comparison of the Lipases Expression in the Myocardium and PVAT

Since various pathological conditions can modulate mRNA and protein expression of the studied lipases, the right atrial appendage and perivascular adipose tissue samples from the patients without CAD and diabetes were used for between-tissue comparison. We observed that the human atrial myocardium of the control patients (NCAD(−)) had approximately 2 times greater mRNA expression and 6 times greater protein level of EL than in perivascular adipose tissue (Figure 6). In addition, the endothelium of the right atrial appendage was characterized by a higher mRNA content of LPL (+2.8 fold) and its significantly higher (over 10 times) protein level than perivascular adipose tissue (Figure 6). The above could be attributed to the greater metabolic activity of the heart when compared to the adipose tissue. This further translates into a higher number of capillaries per area (cap/mm^2^) in the muscle compared to the fat. The human endocardium has around 645 ± 179 capillaries per 1 mm^2^ [57], whereas the capillary density of the left atrial appendage is around 989 ± 173/mm^2^ [58]. In the available literature, we were unable to find the numbers for capillary density in PVAT or even human adipose tissue in general. However, in rats, the capillary density in mesenteric adipose tissue, which is representative of visceral adipose tissue, is approximately 0.007 per mm^2^ [59]. Greater capillary density in the myocardium in turn should translate into a higher expression of EL, which is synthesized mainly by vascular endothelial cells. This is in line with the research by Vigelsø et al. The authors found that although EL is not found in myocytes alone, the density of vascular beds in skeletal muscle positively correlates with EL tissue expression [52].

## 4. Materials and Methods

### 4.1. Patients

The study was performed in accordance with the Guidelines for Good Clinical Practice and the Declaration of Helsinki and approved by the Bioethics Committee of the Medical University of Bialystok (permission number: R-I-002/387/2019). Informed consent was obtained from all subjects involved in the study. The patients were treated at the Department of Cardiosurgery at the University Hospital in Bialystok and classified into four subgroups:Patients qualified for mitral or aortic valve replacement without angiographically confirmed coronary artery disease and without diagnosed type 2 diabetes mellitus (n = 28 patients, designation of the group: NCAD(−)).Patients qualified for mitral or aortic valve replacement without angiographically confirmed coronary artery disease and diagnosed with type 2 diabetes mellitus (n = 6 patients, designation of the group: NCAD(+)).Patients with multivessel coronary artery disease confirmed angiographically and qualified for coronary artery bypass grafting without diagnosed type 2 diabetes mellitus (n = 45 patients, designation of the group: CAD(−)).Patients with multivessel coronary artery disease confirmed angiographically and qualified for coronary artery bypass grafting with diagnosed type 2 diabetes mellitus (n = 23 patients, designation of the group: CAD(+)).

Right atrial appendage samples were collected during right coronary artery cannulation at the time of elective coronary bypass grafting or valve replacement, while perivascular adipose tissue samples were taken from around the right coronary artery in proximity to its aperture. Next, the samples were blotted dry, frozen in liquid nitrogen and stored at −80 °C prior to further analysis. The clinical and laboratory characteristics of the patients are given in Table 1.

### 4.2. Blood Collection

All samples were taken after an 8 h overnight fast. Blood samples were centrifuged for 10 min at 4000 rpm. The collected material was stored at −80 °C until appropriate determinations were made.

### 4.3. Determination of the Apolipoprotein Concentrations

Plasma apolipoprotein levels were detected using Bio-Plex Pro Human Apolipoprotein 10-Plex Assay (Bio-Rad Laboratories, Inc.; Hercules, CA, USA). The assay is a sensitive, magnetic-bead-based multiplex that accurately measures nine apolipoproteins, i.e., Apo A1, Apo A2, Apo B, Apo C1, Apo C3, Apo D, Apo E, Apo H, Apo J and C-reactive protein (CRP). Briefly, plasma samples were centrifuged at 1000 rpm for 15 min at 4 °C to clear the samples of precipitate and immediately diluted (1:50,000). Thereafter, 30 µL of the standards, controls, blanks and samples were added to the appropriate wells of the plate. In the next step, 10 µL of the coupled magnetic beads were added to each well, and the plate was incubated on a shaker at 850 ± 50 rpm for 1 h at room temperature. Subsequently, the plate was washed three times with 1 x assay buffer. The detection antibodies (40 µL) were added to each well, and the plate was incubated again for 1 h. Then 20 µL of the diluted streptavidin–phycoerythrin (SA–PE) was added to each well, and the plate was again incubated and washed three times. Finally, 100 µL of assay buffer was added to each well, and the plate was incubated for 30 s to resuspend the beads. The apolipoprotein concentrations were calculated with consideration of the appropriate standard curve.

### 4.4. ELISA

Plasma concentrations of endothelial lipase and lipoprotein lipase were determined using a commercially available Enzyme-Linked Immunosorbent Assay Kit (ELISA). All the procedures were performed following the manufacturer’s instructions (Cloud-Clone Corp., Houston, TX, USA). The absorbance of assessed lipases was measured spectrophotometrically at 450 nm by the use of a microplate reader (Synergy H1 Hybrid Reader, BioTek Instruments, Winooski, VT, USA). The EL and LPL concentrations were calculated from the obtained standard curves, while the results were expressed in nanograms per milliliter of the plasma.

### 4.5. Quantitative Real-Time PCR

Total RNA was isolated according to the procedure described previously [60]. Briefly, the NucleoSpin RNA Plus Kit (Macherey Nagel GmbH & Co.KG, Duren, Germany) was used for the RNA purification, while traces of genomic DNA were excluded by DNase treatment (Ambion, Thermo Fisher Scientific, Waltham, MA, USA). Furthermore, the quality and quantity of extracted RNA were assessed by spectrophotometric measurements (at an absorbance OD ratio of 260/280 and 260/230). Reverse transcription was performed using the EvoScript universal cDNA master kit (Roche Molecular Systems, Boston, MA, USA), whereas amplification of the product was performed using the FastStart essential DNA green master (Roche Molecular Systems) in a LightCycler 96 System Real-Time thermal cycler. The following reaction parameters were applied: 15 s denaturation at 94 °C, 15 s annealing at 55 °C for RPL13A, 62 °C for EL and 60 °C for LPL, and then a 15 s extension at 72 °C for 45 cycles. The primer sequences: *RPL13A* sense 5′-CTATGACCAATAGGAAGAGCAACC-3′, antisense 5′-GCAGAGTATATGACCAGGTGGAA-3′; *EL* sense 5′-GGCCACATTGACATCTACCC-3′, antisense 5′-ACGGCTCGCTCATGCTCACA-3′; *LPL* sense 5′-GAGATTTCTCTGTATGGCACC-3′, antisense 5′-CTGCAAATGAGACACTTTCTC-3′. The primers’ efficiency was analyzed using the standard curve method. Finally, the gene expression was calculated according to the Pfaffl method [61] normalizing to the housekeeper gene (RPL13A).

### 4.6. Protein Extraction and Western Blot

The protein content of the studied lipases was detected using a routine Western blotting procedure. Tissues were homogenized in RIPA buffer containing protease and phosphatase inhibitors (Roche Diagnostics GmbH, Mannheim, Germany), and the protein concentration in the homogenates was measured using the BCA method with bovine serum albumin (fatty acid-free, Sigma–Aldrich, St. Louis, MO, USA) as a standard. Then, homogenates were reconstituted in Laemmli buffer (Bio-Rad, Hercules, CA, USA), and equal amounts of the proteins (20 µg per sample) were loaded on Criterion TGX Stain-Free Precast Gels (Bio-Rad, Hercules, CA, USA). Subsequently, the separated proteins were transferred onto polyvinylidene fluoride (PVDF) membranes and after blocking in 5% non-fat dry milk for 1 h, the membranes were incubated overnight at 4 °C with the primary antibodies, i.e., EL (Abcam, Cambridge, United Kingdom), LPL and GAPDH (Santa Cruz Biotechnology, Inc., Dallas, TX, USA). The membranes were then incubated with suitable anti-rabbit or anti-mouse IgG horseradish peroxidase-conjugated secondary antibodies (Santa Cruz Biotechnology, Dallas, TX, USA). Finally, the protein bands were imaged by chemiluminescence using Clarity Western ECL Substrate (Bio-Rad, Hercules, CA, USA) and quantified densitometrically via ChemiDoc visualization system (Image Laboratory Software Version 6.0.1; Bio-Rad, Hercules, CA, USA). The protein expression (optical density arbitrary units) was normalized to GAPDH reference protein expression.

### 4.7. Statistics

The data obtained in this study were pre-processed using MS Excel. In the following step, data underwent statistical analysis using R (Version 3.6.3). To that end, they were tested for normality of distribution (Shapiro–Wilk test) and equal variances (Fligner-Kileen test). Whenever the above criteria were fulfilled, one-way ANOVA followed by pairwise Student’s *t*-tests were applied in order to detect the between-group differences. In case the prerequisites were not met, the Kruskal–Wallis H test followed by pairwise Wilcoxon’s tests were used. To control for false positives, the obtained *p*-values were adjusted using Benjamini and Hochberg (BH) correction. The adjusted *p*-values ≤ 0.05 were considered to be statistically significant. For consistency, all the between-group comparisons were presented graphically as boxplots. To test the dependence between the variables of interest, Pearson correlation coefficients were applied. The data were analyzed with (NCAD(−), NCAD(+), CAD(−), CAD(+)) and without separations to experimental groups (all groups together). To counteract the occurrence of false-positive correlations, the *p*-values obtained in this analysis were adjusted using the aforementioned Benjamini and Hochberg procedure. The correlations were depicted graphically as heat maps.

## 5. Conclusions

In our study, a decreased level of ApoA1 and an elevated plasma concentration of ApoC1 and apoH were associated with the presence of CAD. These findings suggest that these apolipoproteins may serve as potential biomarkers useful to identify subjects at risk of developing CAD. In addition, CAD had a relatively small effect on the plasma concentration of endothelial and lipoprotein lipases. However, additional research is needed to better understand the relationship between these markers (i.e., apolipoproteins, EL and LPL) and CAD, as well as to explore their potential utility in clinical practice. On the opposite, the presence of diabetes in non-CAD patients significantly increased plasma concentration and transcript content of both lipases in the perivascular adipose tissue. Moreover, the human atrial myocardium has significantly higher expression of EL protein (approximately 5-fold) and LPL (over 10-fold) than the perivascular adipose tissue. The results are consistent with a high metabolic rate and vascularization of the heart.

## 6. Limitations

Our present study, although interesting, is not without limitations, which must include: (1) the relatively advanced age of the patients (>60 years of age); (2) slight differences with respect to gender distribution between the groups and overall had a prevalence of male subjects; (3) the accompanying medication taken by the patients (statins); (4) low number (n = 6) of patients in the NCAD(+) group. The above-mentioned factors could potentially impact the power of statistical tests performed and obfuscate the results. Unfortunately, there was little that could be performed to counteract them. Clinical practice demonstrates that the co-occurrence of the conditions (absence of CAD, presence of DM, recommended mitral or aortic valve replacement) is very rare.

## Figures and Tables

**Figure 1 ijms-24-13552-f001:**
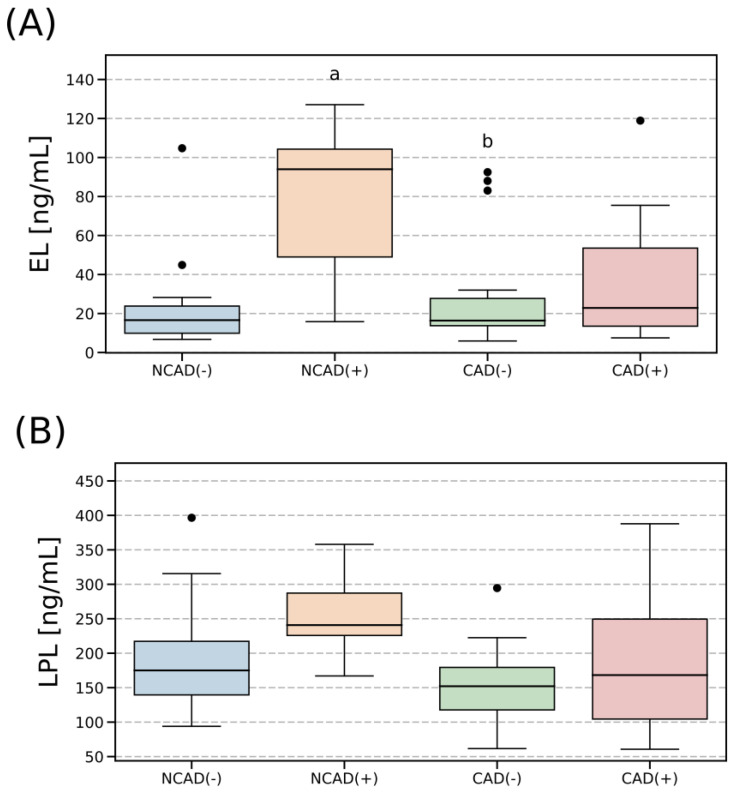
Plasma endothelial lipase (**A**) and lipoprotein lipase (**B**) concentration in CAD and non-CAD patients. The inner horizontal line of a box represents the median. Box boundaries: 25–75 percentile, box whiskers: 1.5 interquartile range (IQR) or max/min value in the group. Solid black dot—data point outside 1.5 IQR. The values are expressed in nanograms per milliliter of plasma. a—difference vs. NCAD(−); b—difference vs. NCAD(+); *p* < 0.05. CAD(−)—patients with coronary artery disease and without diabetes mellitus; CAD(+)—patients with coronary artery disease and with diabetes mellitus; EL—endothelial lipase; LPL—lipoprotein lipase; NCAD(−)—patients without coronary artery disease and without diabetes mellitus; NCAD(+)—patients without coronary artery disease and with diabetes mellitus.

**Figure 2 ijms-24-13552-f002:**
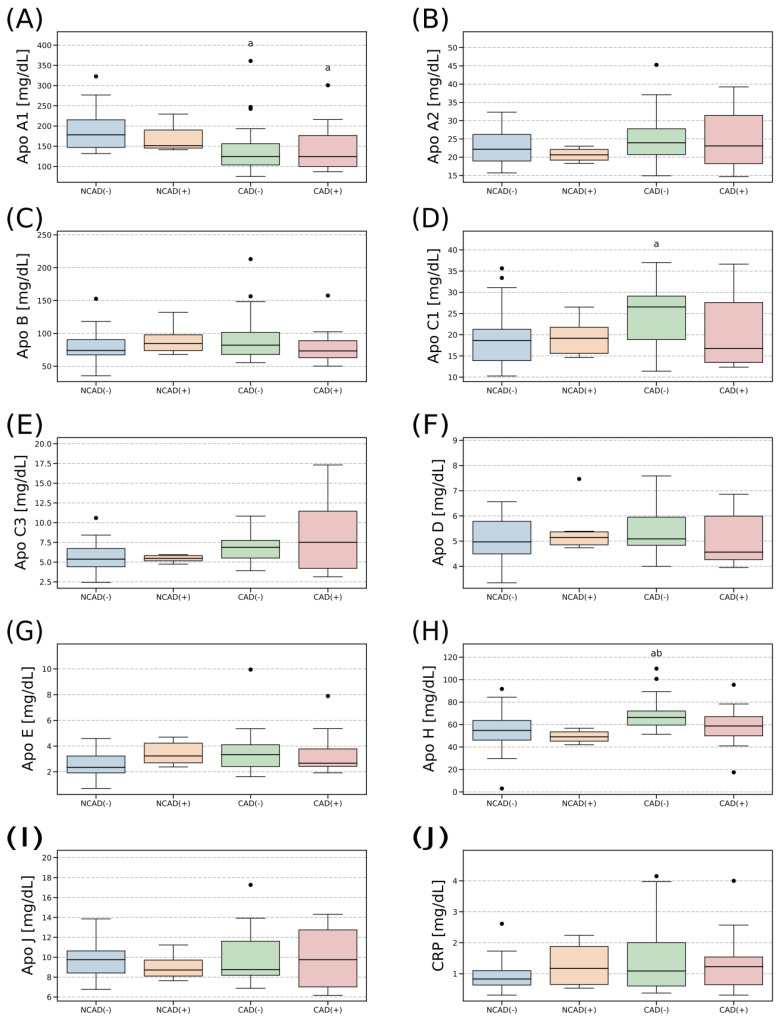
Plasma apolipoproteins: ApoA1 (**A**), ApoA2 (**B**), ApoB (**C**), ApoC1 (**D**), ApoC3 (**E**), ApoD (**F**), ApoE (**G**), ApoH (**H**), ApoJ (**I**) and CRP (**J**) concentrations in CAD and non-CAD patients. The inner horizontal line of a box represents the median. Box boundaries: 25–75 percentile, box whiskers: 1.5 interquartile range (IQR) or max/min value in the group. Solid black dot—data point outside 1.5 IQR. The values are expressed in milligrams per deciliter of plasma. a—difference vs. NCAD(−); b—difference vs. NCAD(+); *p* < 0.05. CAD(−)—patients with coronary artery disease and without diabetes mellitus; CAD(+)—patients with coronary artery disease and with diabetes mellitus; NCAD(−)—patients without coronary artery disease and without diabetes mellitus; NCAD(+)—patients without coronary artery disease and with diabetes mellitus.

**Figure 3 ijms-24-13552-f003:**
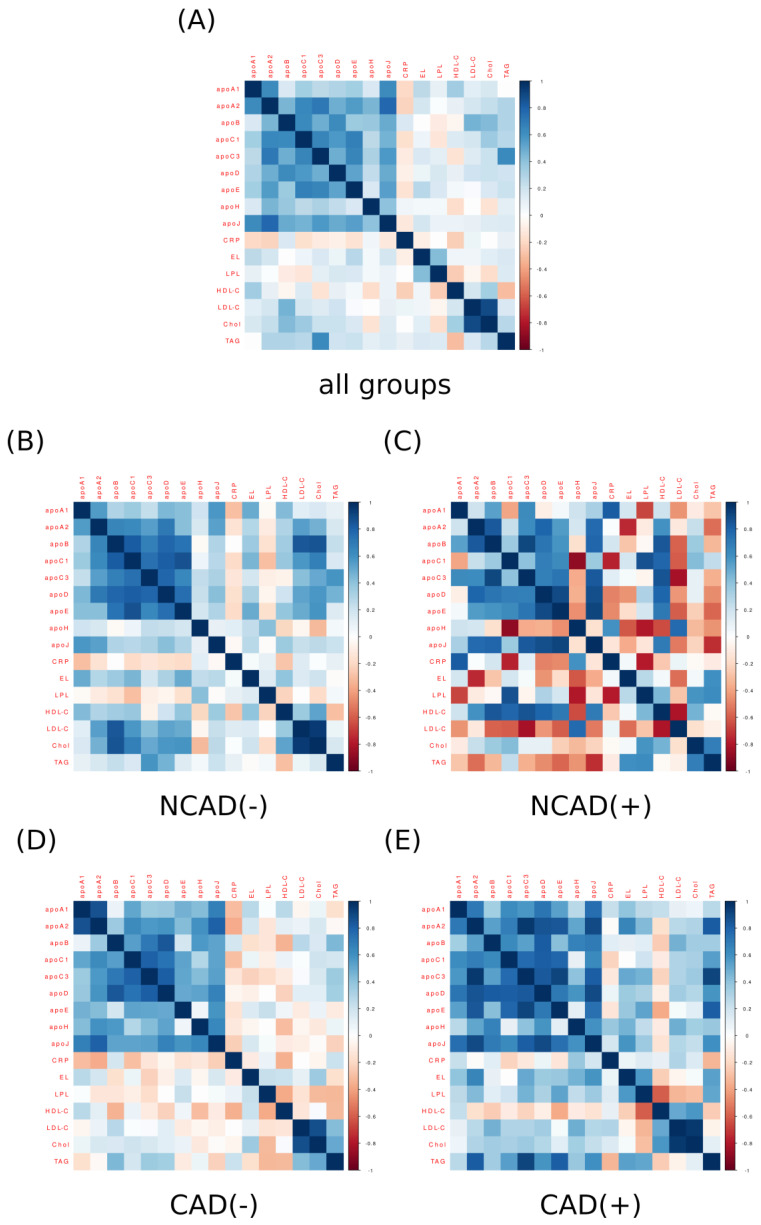
Correlation matrix (heatmap) for plasma concentration of lipases, apolipoproteins and blood lipid profile (Chol, HDL-C, LDL-C and TAG) in all the studied groups together (**A**) and in each group separately: NCAD(−) (**B**), NCAD(+) (**C**), CAD(−) (**D**) and CAD(+) (**E**). Pearson correlation coefficients are depicted as shades of blue (positive correlation) or red (negative correlation). CAD(−)—patients with coronary artery disease and without diabetes mellitus; CAD(+)—patients with coronary artery disease and with diabetes mellitus; Chol—total cholesterol; HDL-C—high-density lipoprotein cholesterol; LDL-C—low-density lipoprotein cholesterol; NCAD(−)—patients without coronary artery disease and without diabetes mellitus; NCAD(+)—patients without coronary artery disease and with diabetes mellitus; TAG—triacylglycerol.

**Figure 4 ijms-24-13552-f004:**
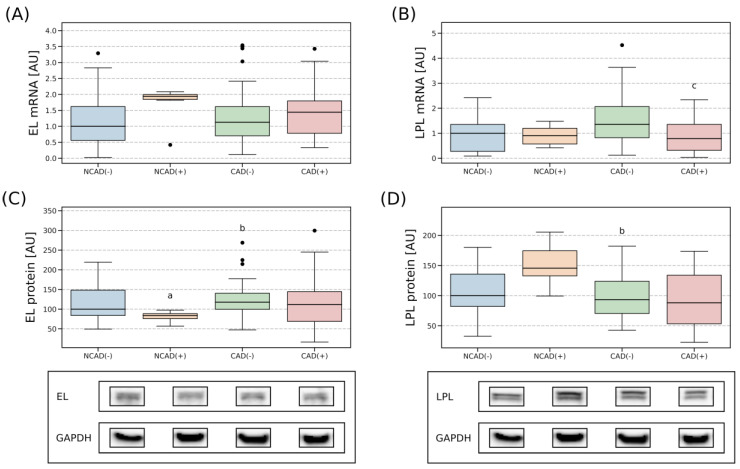
Gene and protein expression of endothelial lipase (**A**,**C**) and lipoprotein lipase (**B**,**D**) in the vascular beds of the right atrial appendage in CAD and non-CAD patients. The inner horizontal line of a box represents the median. Box boundaries: 25–75 percentile, box whiskers: 1.5 interquartile range (IQR) or max/min value in the group. Solid black dot—data point outside 1.5 IQR. Representative bands of Western Blot analysis were shown. a—difference vs. NCAD(−); b—difference vs. NCAD(+); c—difference vs. CAD(−), *p* < 0.05. CAD(−)—patients with coronary artery disease and without diabetes mellitus; CAD(+)—patients with coronary artery disease and with diabetes mellitus; EL—endothelial lipase; LPL—lipoprotein lipase; NCAD(−)—patients without coronary artery disease and without diabetes mellitus; NCAD(+)—patients without coronary artery disease and with diabetes mellitus.

**Figure 5 ijms-24-13552-f005:**
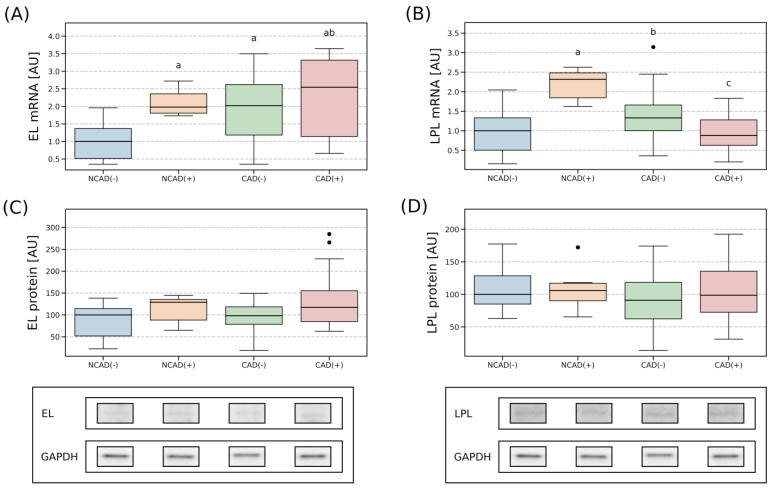
Gene and protein expression of endothelial lipase (**A**,**C**) and lipoprotein lipase (**B**,**D**) in the vascular beds of perivascular adipose tissue in CAD and non-CAD patients. The inner horizontal line of a box represents the median. Box boundaries: 25–75 percentile, box whiskers: 1.5 interquartile range (IQR) or max/min value in the group. Solid black dot—data point outside 1.5 IQR. Representative bands of Western Blot analysis were shown. a—difference vs. NCAD(−); b—difference vs. NCAD(+); c—difference vs. CAD(−), *p* < 0.05. CAD(−)—patients with coronary artery disease and without diabetes mellitus; CAD(+)—patients with coronary artery disease and with diabetes mellitus; EL—endothelial lipase; LPL—lipoprotein lipase; NCAD(−)—patients without coronary artery disease and without diabetes mellitus; NCAD(+)—patients without coronary artery disease and with diabetes mellitus.

**Figure 6 ijms-24-13552-f006:**
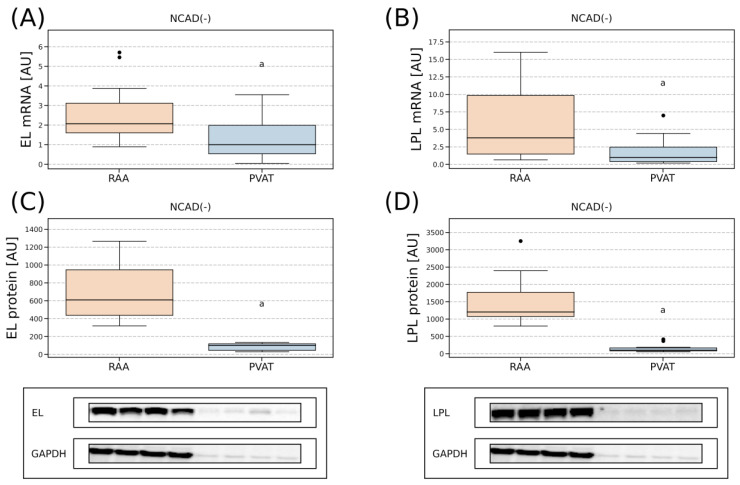
Between tissue comparison of gene and protein expression pattern for endothelial lipase (**A**,**C**) and lipoprotein lipase (**B**,**D**) in the right atrial appendage and perivascular adipose tissue of the control patients (NCAD(−)). The gene expression of the lipases was normalized to the housekeeper gene (RPL13A), whereas protein expression was normalized to the GAPDH protein level. The inner horizontal line of a box represents the median. Box boundaries: 25–75 percentile, box whiskers: 1.5 interquartile range (IQR) or max/min value in the group. Solid black dot—data point outside 1.5 IQR. Representative bands of Western blot analysis were shown. a—difference vs. EL—endothelial lipase; LPL—lipoprotein lipase; NCAD(−)—patients without coronary artery disease and without diabetes mellitus; PVAT—perivascular adipose tissue; RAA—right atrial appendage.

**Figure 7 ijms-24-13552-f007:**
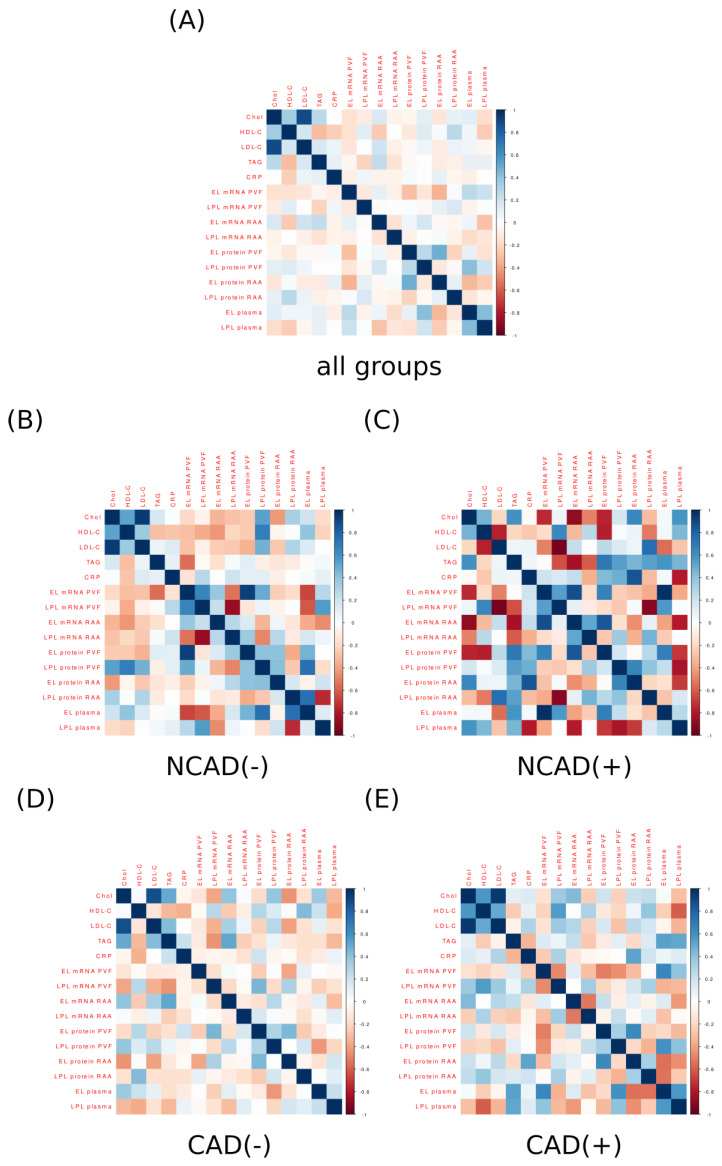
Correlation matrix (heatmap) of lipases (in plasma and studied tissues) and blood lipid profile (Chol, HDL-C, LDL-C, TAG) in all the studied groups together (**A**) and in each group separately: NCAD(−) (**B**), NCAD(+) (**C**), CAD(−) (**D**) and CAD(+) (**E**). Pearson correlation coefficients are depicted as shades of blue (positive correlation) or red (negative correlation). CAD(−)—patients with coronary artery disease and without diabetes mellitus; CAD(+)—patients with coronary artery disease and with diabetes mellitus; Chol—total cholesterol; EL—endothelial lipase; HDL-C—high-density lipoprotein cholesterol; LDL-C—low-density lipoprotein cholesterol; LPL—lipoprotein lipase; NCAD(−)—patients without coronary artery disease and without diabetes mellitus; NCAD(+)—patients without coronary artery disease and with diabetes mellitus; PVAT—perivascular adipose tissue; RAA—right atrial appendage; TAG—triacylglycerol.

**Table 1 ijms-24-13552-t001:** Clinical characteristics of the CAD and non-CAD patients.

	NCAD(−)	NCAD(+)	CAD(−)	CAD(+)
**Age (years)**	61.6 ± 8.52	65.3 ± 7.12	63.6 ± 8.1	67.3 ± 7.95
**Gender (female/male)**	9/19	3/3	5/40	8/15 bc
**DM (no/yes)**	28/0	0/6 a	45/0 b	0/23 ac
**Glucose (mg/dL)**	99.4 ± 16.32	120.3 ± 16.99 a	95.5 ± 14.32	134.4 ± 30.18 ac
**GFR (mL/min)**	93.0 ± 16.26	76.3 ± 23.69	88.7 ± 18.55	78.5 ± 18.63
**Total cholesterol (mg/dL)**	193.0 ± 42.69	172.3 ± 21.8	183.4 ± 40.55	169.4 ± 47.58
**HDL-C (mg/dL)**	52.9 ± 10.81	42.2 ± 11.00	46.6 ± 11.09	41.4 ± 8.83 a
**LDL-C (mg/dL)**	116.9 ± 37.01	117.5 ± 11.69	108.3 ± 36.73	105.4 ± 48.68
**TAG (mg/dL)**	126.0 ± 56.4	114.0 ± 40.71	146.3 ± 82.52	163.4 ± 91.65
**CRP (mg/dL)**	5.2 ± 5.6	3.5 ± 2.6	5.1 ± 9.34	4.2 ± 4.26
**EF (%)**	52.4 ± 6.88	43.0 ± 11.78	53.0 ± 8.63	52.8 ± 8.99
**BMI (kg/m^2^)**	28.5 ± 4.62	29.4 ± 5.34	27.4 ± 3.31	29.9 ± 4.57
**Hb (g/dL)**	13.9 ± 1.29	13.4 ± 1.48	13.6 ± 1.42	13.6 ± 1.34
**MI (no/yes)**	28/0	6/0	28/17 a	16/7 a

Data are presented as the mean and standard deviation. *p* < 0.05; a—difference vs. NCAD(−); b—difference vs. NCAD(+); c—difference vs. CAD(−). BMI—body mass index; CAD(−)—patients with coronary artery disease and without diabetes mellitus; CAD(+)—patients with coronary artery disease and with diabetes mellitus; CRP—C-reactive protein; DM—diabetes mellitus; EF—ejection fraction; GFR—glomerular filtration rate; Hb—hemoglobin; HDL-C—high-density lipoprotein cholesterol; LDL-C—low-density lipoprotein cholesterol; MI—myocardial infraction; NCAD(−)—patients without coronary artery disease and without diabetes mellitus; NCAD(+)—patients without coronary artery disease and with diabetes mellitus; TAG—triacylglycerol.

## Data Availability

The data presented in this study are available upon request.

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
