# Peer review of "Multivessel Coronary Artery Disease Complicated by Diabetes Mellitus Has a Relatively Small Effect on Endothelial and Lipoprotein Lipases Expression in the Human Atrial Myocardium and Coronary Perivascular Adipose Tissue"

_ijms, 2023, doi:10.3390/ijms241713552_

Round 1

Reviewer 1 Report

Knapp et al analyzed the endothelial and lipoprotein lipases in human atrial myocardium 4 and the coronary perivascular adipose tissue, along with other biomarkers in patients with coronary diseases with and without diabetes mellitus. However, there are only 6 patients in NCAD (+) group, compared with n=23 in CAD(+) group, n=45 in CAD(-) group, and n=28 in NCAD(-) group. This might reduce the statistical power, especially when authors tried to analyze the impact of diabetic mellitus on those markers. Authors should discuss that. For mRNA and protein levels of EL and LPL from PVAT and RAA, the total mRNA and protein in fat tissues (PVAT) might be much less than in myocardium tissues (RAA). Did the authors normalize the mRNA amount in real-time PCR and the protein amount in Western blot analysis between these two types of tissues? They should clearly state this in the method part, and discuss it along with the results. In addition, Figure 6 used the CAD(-) and NCAD(-) samples for EL and LPL analysis (both mRNA level and protein level) between tissues, but the western blot showed all four groups' data, which is not consistent with mRNA data. Plus the EL Western blot band in PVF has five samples, this need to be explained. Overall, the manuscript needs some more revision and clarification. Authors also need carefully proofread the manuscript.

Need the proofread. 

Reviewer 2 Report

The paper is interesting, quite well written and methodological correct.

This reviewer raises few issues that need to be addressed by the authors.

1- An interesting U-shaped correlation has recently been observed between HDL cholesterol and retinal microangiopathy (Diabetes Res Clin Pract. 2019 Apr;150:236-244. doi: 10.1016/j.diabres.2019.03.028) which confirms previous similar observations in atherosclerotic macroangiopathy (lines 79-80). This original observation should be added in discussion for its pathophysiological as well as clinical impact.

2- A paragraph on the limitations of the study is missing. For example, the population studied is over 60 aged , furthermore the male gender clearly prevails over the female one. Therefore the conclusions of the study are not generalizable to other settings. The authors should address these issues in the discussion.

3- Recently, the NID-2 study post-hoc analysis, a multicenter randomized trial, has allowed to demonstrate that following an optimized multifactorial drug treatment, the number of CV risk factors that reach the guideline target correlates with the reduction of the MACEs and overall mortality (Cardiovasc Diabetol. 2022; 21: 235.  doi: 10.1186/s12933-022-01674-7). Thus, the authors should comment above important findings in the discussion.

Minor editing of English language is required.

Round 2

Reviewer 1 Report

The responses from the authors are fine with me, but I still have some questions for Fig. 6. Is the Western blot from Figure 6 the same as Figure 4 (RAA) and 5 (PVAT)’s protein Western blot? Looking at PVAT data, the intensity of bands is different from Figure 6 and those in Fig. 5 and Fig. 4. Further, when the authors claimed the normalization against GAPDH for Western blot data, do they take the ratio of band intensity of the sample against that of GAPDH? For samples from RAA, both GAPDH and the protein of interest are quite dark in RAA, and are quite faint in PVAT, which makes sense, but when taking the ratio, each group seems the same, which is different from what the authors’ results claimed.  

proof reading
